# Adoptive cancer immunotherapy using DNA-demethylated T helper cells as antigen-presenting cells

Alexei F. Kirkin[1,2], Karine N. Dzhandzhugazyan[1,2], Per Guldberg [1], Johnny Jon Fang[1,2], Rikke S. Andersen [3], Christina Dahl [1], Jann Mortensen [4], Tim Lundby[4], Aase Wagner[5], Ian Law[4], Helle Broholm[6], Line Madsen[7], Christer Lundell-Ek [2], Morten F. Gjerstorff [3], Henrik J. Ditzel [3,8], Martin R. Jensen[2] & Walter Fischer[2,9]

In cancer cells, cancer/testis (CT) antigens become epigenetically derepressed through DNA demethylation and constitute attractive targets for cancer immunotherapy. Here we report that activated CD4$^+$ T helper cells treated with a DNA-demethylating agent express a broad repertoire of endogenous CT antigens and can be used as antigen-presenting cells to generate autologous cytotoxic T lymphocytes (CTLs) and natural killer cells. In vitro, activated CTLs induce HLA-restricted lysis of tumor cells of different histological types, as well as cells expressing single CT antigens. In a phase 1 trial of 25 patients with recurrent glioblastoma multiforme, cytotoxic lymphocytes homed to the tumor, with tumor regression ongoing in three patients for 14, 22, and 27 months, respectively. No treatment-related adverse effects were observed. This proof-of-principle study shows that tumor-reactive effector cells can be generated ex vivo by exposure to antigens induced by DNA demethylation, providing a novel, minimally invasive therapeutic strategy for treating cancer.

[1] Danish Cancer Society Research Center, 2100 Copenhagen, Denmark. [2] CytoVac A/S, 2970 Hørsholm, Denmark. [3] Department of Cancer and Inflammation Research, Institute for Molecular Medicine, University of Southern Denmark, 5000 Odense, Denmark. [4] Department of Clinical Physiology, Nuclear Medicine and PET, Copenhagen University Hospital, Rigshospitalet, 2100 Copenhagen, Denmark. [5] Department of Neuroradiology, Copenhagen University Hospital, Rigshospitalet, 2100 Copenhagen, Denmark. [6] Department of Neuropathology, Copenhagen University Hospital, Rigshospitalet, 2100 Copenhagen, Denmark. [7] Department of Pathology, Aarhus University Hospital, 8000 Aarhus, Denmark. [8] Department of Oncology, Odense University Hospital, 5000 Odense, Denmark. [9] Department of Neurosurgery, Copenhagen University Hospital, Rigshospitalet, 2100 Copenhagen, Denmark. Alexei F. Kirkin and Karine N. Dzhandzhugazyan contributed equally to this work. Correspondence and requests for materials should be addressed to A.F.K. (email: aki@cancer.dk or aki@cytovac.dk)

Adoptive transfer of naturally occurring or genetically engineered immune effector cells has demonstrated therapeutic benefit in clinical trials of advanced cancers[1, 2]. One successful approach is the adoptive transfer of autologous tumor-infiltrating lymphocytes (TILs) in melanoma patients resulting in complete response rates of up to 40%[3]. Alternative approaches utilize T cells genetically engineered to confer specificity for tumor-associated antigens by introducing a cloned T cell receptor (TCR) or a chimeric antigen receptor (CAR)[2]. Early phase clinical trials of these strategies have yielded promising results in the treatment of melanoma and other cancers[4, 5].

A critical determinant of tumor eradication by adoptive immunotherapy is the tumor-associated antigen(s) recognized by cytotoxic T lymphocytes (CTLs). One major class of cancer rejection antigens encompasses neoantigens, which arise through tumor-specific DNA alterations that lead to the generation of aberrant proteins[6]. Neoantigens usually differ from patient to patient, and are thought to be the major targets in TIL-based therapies and therapies aiming at nonspecific immune activation through inhibition of T cell checkpoint proteins, such as CTLA-4 and PD-1. The second major class of cancer rejection antigens encompasses cancer/testis (CT) antigens (also known as cancer germline antigens), a heterogeneous group of >100 proteins of different families with largely unknown functions[7]. CT antigens are repressed in normal adult tissues, with the exception of non-major histocompatibility complex (MHC)-expressing germ cells, but are aberrantly re-expressed in most human cancers due to promoter demethylation[7, 8]. Clinical trials utilizing T cells genetically engineered to recognize single CT antigens, such as MAGE-A3 or CTAG1 (also known as NY-ESO-1), have shown high response rates for selected patient groups[4, 9], but this approach generally has limitations due to extensive interpatient and intratumor heterogeneity of CT-antigen expression[7].

Herein we describe an autologous procedure developed to induce an immune response against a broad repertoire of CT antigens, the key elements of which are (1) generation of proliferating activated $CD4^+$ T "helper" ($T_H$) cells by incubation of normal peripheral blood lymphocytes (PBLs) with fully mature dendritic cells (DCs); (2) induction of endogenous CT-antigen expression in activated $T_H$ cells by treatment with a DNA-demethylating agent, and (3) ex vivo immunization of normal lymphocytes using demethylated $T_H$ cells as antigen-presenting cells. The CTLs and natural killer (NK) cells generated by this procedure exhibit early differentiation phenotypes, both expressing CD62L (also known as L-selectin), and can potentially be used for treatment of a broad range of advanced human cancers. We have tested this approach in a phase 1 clinical trial (NCT01588769) of patients with late-stage recurrent glioblastoma multiforme (GBM), a highly malignant primary brain tumor usually associated with a rapidly fatal clinical course[10, 11].

## Results

**Generation of $T_H$ cell-enriched lymphocyte populations.** Gene repression by DNA cytosine methylation may be reversed by the action of nucleoside-based inhibitors of DNA methyltransferase. However, as DNA replication is required for this process[12], drug-induced demethylation is not feasible in non-dividing antigen-presenting cells such as DCs. Instead, we focused our work on $T_H$ cells, which also can function as antigen-presenting cells for generation of autologous CTLs[13]. Proliferation of isolated $T_H$ cells can be effectively stimulated by incubation with phytohemagglutinin (PHA)[13] or a combination of antibodies against CD3 and CD28[14]. However, to avoid the possible adverse effects associated with the use of foreign proteins for immunization

procedures[15], we exploited our initial observation that co-culturing unseparated PBLs with autologous fully mature antigen-unloaded DCs induced intense lymphocyte proliferation and enrichment of $T_H$ cells (Fig. 1).

The DCs used for this activation and enrichment procedure were matured in medium containing IL-1β, IL-6, TNF-α, and PGE2. They expressed high levels of CD40, CD80, and CD86 (Fig. 1a) and, during maturation, released the Th1-polarizing cytokine, IL-12p70 ($200.6 \pm 84.1$ pg/$10^6$ cells; seven independent cultures). Incubation of DCs with autologous lymphocytes in the presence of low concentrations of IL-2 (25 IU/ml) resulted in intensive cell proliferation after ~7 days (Fig. 1b). The most rapidly proliferating were $CD4^+$ cells, which constituted ~50% in the parental cultures and >80% in the DC-induced cultures (Fig. 1c). Interestingly, these $CD4^+$ cells were highly positive for HLA class II, CD80 and CD86, all markers of professional antigen-presenting cells[16], and also expressed CD70, a ligand for CD27 and marker of Th1 helper cells[17] (Fig. 1d). There was only minimal expression of stress-associated molecules (Supplementary Fig. 1). Of note, no proliferation of lymphocyte cultures was observed in the absence of DCs, even in the presence of IL-2 (Fig. 1b). In contrast to stimulation with DCs, treatment of PBLs with PHA did not induce preferential proliferation of $CD4^+$ cells (Supplementary Table 1).

**Epigenetic derepression of CT antigens.** Having established conditions for generating lymphocyte cultures enriched for proliferating $T_H$ cells, we next attempted to induce expression of endogenous CT antigens in these cells by epigenetic manipulation. In cancer cell lines, derepression of genes silenced by promoter hypermethylation, including CT antigens[18], can be readily induced in vitro by treatment with demethylating agents such as 5-aza-2′-deoxycytidine (5-aza-CdR). Although it has proven considerably more difficult to induce a state of DNA demethylation in normal human cells[18], a few studies have demonstrated 5-aza-CdR-induced re-expression of some MAGE antigens in primary fibroblasts and PHA-activated lymphocytes[19, 20].

To investigate whether CT antigens can be derepressed in DC-induced lymphocytes, we tested a number of 5-aza-CdR concentrations, culture conditions and incubation times. Robust and consistent expression of selected CT antigens representing different families (MAGE, GAGE, CTCFL, and CTAG1) was achieved by treating cells in logarithmic growth phase with 10 μM 5-aza-CdR for 2–3 days in the presence of IL-2 (150 IU/ml). The phenotype of 5-aza-CdR-treated cells was similar to that of untreated cells (Fig. 2a, b), and only a slight induction in the expression of the stress proteins MICA/MICB was observed (Supplementary Fig. 1). Reverse transcription polymerase chain reaction (RT-PCR) analysis of selected CT antigens showed complete lack of expression in untreated lymphocytes, but clear induction of expression in 5-aza-CdR-treated cells (Fig. 2c and Supplementary Figs. 2 and 3). After termination of 5-aza-CdR treatment, expression of CT antigens remained stable for at least 3 days (Supplementary Fig. 4). Analysis of separated $CD4^+$ and $CD8^+$ T-cell populations showed that expression of CT antigens was induced by 5-aza-CdR in both cell types at comparable levels (Supplementary Fig. 5).

MAGE expression was also demonstrated at the protein level using the antibody 57B, which recognizes an epitope common to several MAGE antigens[21]. The specificity of the 57B antibody was confirmed by sandwich ELISA (Supplementary Fig. 6). Consistent with the data on mRNA expression, MAGE proteins were clearly detectable in 5-aza-CdR-treated lymphocytes by immunoblotting (Fig. 2d and Supplementary Fig. 7) and cell-based ELISA (Fig. 2e), albeit at lower levels than in MDA-MB-231 breast cancer cells.

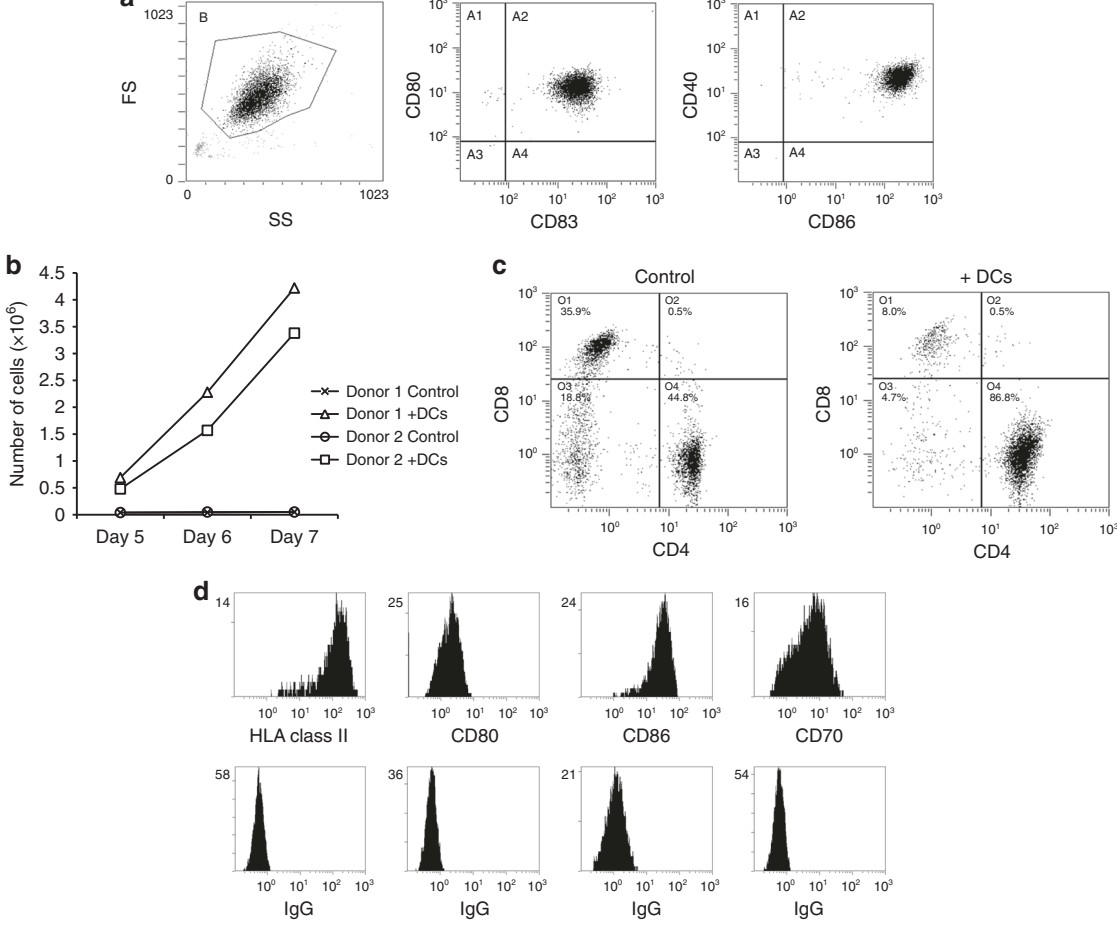

**Fig. 1** Fully mature DCs induce lymphocyte proliferation and enrichment for $T_H$ cells. **a** Flow-cytometric analysis of the expression of CD40, CD80, CD83 and CD86 on the DCs used for stimulation of lymphocyte proliferation. **b** PBLs from two healthy donors cultured in the presence or absence of fully mature DCs. Shown are total numbers of large lymphocytes (>9 μm). **c** Flow-cytometric analysis of CD4 and CD8 expression in lymphocyte cultures grown for 6 days in the presence or absence of fully mature DCs. **d** Flow-cytometric analysis of the expression of the indicated markers on DC-induced lymphocytes

Changes in DNA methylation were evaluated in lymphocytes after 1–3 days of 5-aza-CdR treatment. Quantitative analysis of the methylation status of CpG sites in the promoter regions of CT-antigen genes showed a gradual decrease in methylation density with increasing time of incubation (Fig. 2f and Supplementary Fig. 8). These data support the theory that 5-aza-CdR-induced derepression of CT antigens is mediated via demethylation of the corresponding gene promoters.

**Generation of CTL and NK-cell responses**. With the successful derepression of CT antigens in $T_H$-enriched populations of activated lymphocytes, we next used these cells as antigen-presenting cells for in vitro immunization. Incubation of autologous PBLs with 5-aza-CdR-treated cells led to intensive cell proliferation after 9-11 days. The resulting cell cultures consisted of two dominant subpopulations; NK (CD56+ CD3−) and CD3+ T cells (predominantly CD8+ T cells) (Fig. 3a). Interestingly, both subpopulations were characterized by a high expression of CD62L, and a significant proportion of CD3+ T cells expressed CD27 (Fig. 3a). The CD3+ T cells expressed CD45RO and predominantly αβ-TcR (relative to γδ-TcR; Supplementary Table 2). A large proportion of the NK cells expressed CD16 (Supplementary Table 2). Cultivation of 5-aza-CdR-treated cells alone for 11 days did not result in an increase in cell number, and

autologous PBLs grown in the absence of 5-aza-CdR-treated cells significantly decreased in number (Supplementary Fig. 9).

To investigate whether the immunization procedure resulted in clonal expansion of T cells, we employed comparative TCR clonotype mapping. This technique utilizes the fact that clonotypic TCR transcripts have no junctional diversity and therefore can be resolved as distinct bands by denaturing gradient gel electrophoresis (DGGE)[22]. Analysis of untreated PBLs from donors rarely revealed distinct bands, consistent with a polyclonal TCR repertoire. In contrast, analysis of lymphocytes expanded upon co-culture with 5-aza-CdR-treated cells showed the presence of multiple clonotypic TCR transcripts, ranging from 2 to >10 for each BV family (Fig. 3b).

Both NK cells and T cells can induce lysis of tumor cells, but via different mechanisms. Lysis by T cells requires recognition of an antigenic peptide-MHC class I complex on the surface of target cells and is usually associated with the production of cytokines such as IFN-γ and TNF-α. Unseparated cytotoxic lymphocytes did not interact with untreated autologous $T_H$ cells, whereas they did recognize 5-aza-CdR-treated $T_H$ cells (Supplementary Fig. 10A). To test the activity of the T-cell component of cytotoxic lymphocytes prepared from the blood of healthy donors, we incubated CD56-depleted cells (hereinafter referred to as CTLs) with autologous 5-aza-CdR-treated $T_H$ cells. As shown in Supplementary Fig. 10B, this led to an increase in the

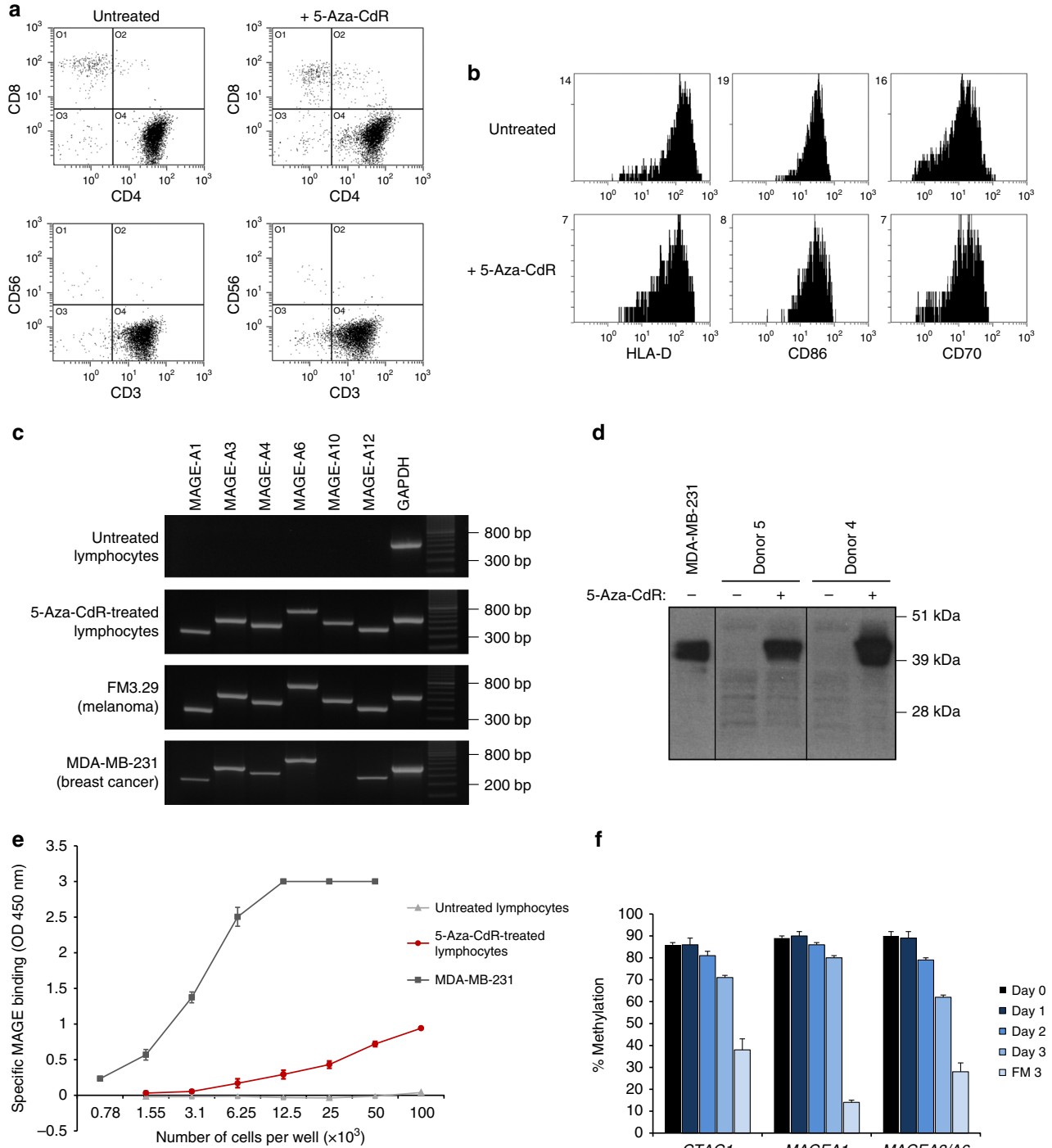

**Fig. 2** Treatment of DC-induced lymphocytes with 5-aza-CdR induces derepression of CT antigens. **a** Phenotype of DC-induced lymphocytes before and after treatment with 10 μM 5-aza-CdR for 2 days. **b** Flow-cytometric analysis of the expression of the indicated markers on DC-induced lymphocytes grown in the presence or absence of 5-aza-CdR. **c** Analysis of MAGE mRNA expression by RT-PCR in DC-activated lymphocytes cultured in the presence or absence of 10 μM 5-aza-CdR for 2 days. GAPDH expression was analyzed as a control. MDA-MB-231 breast cancer cells and FM3.29 melanoma cells were included as examples of MAGE expression in human cancer cells. **d** Analysis of MAGE protein expression by immunoblotting. Whole-cell lysates from untreated lymphocytes, 5-aza-CdR-treated lymphocytes and MDA-MB-231 cells were analyzed using the 57B anti-MAGE antibody. Total protein loaded: Lane 1: 4.3 μg; lanes 2–5: 30-36 μg. **e** Cell-based ELISA. Cells were plated in four series of seven 2-fold dilutions. The difference in optical density between the mean of two readings with the 57B antibody and the mean of two readings with control antibodies was plotted as specific binding ± s.d. The limit of detection of the ELISA reader was 3.0. **f** Average methylation levels at CT-antigen gene promoters measured by bisulfite pyrosequencing in lymphocytes treated with 10 μM 5-aza-CdR for 0–3 days. Data of triplicates from one experiment are represented as means ± s.d

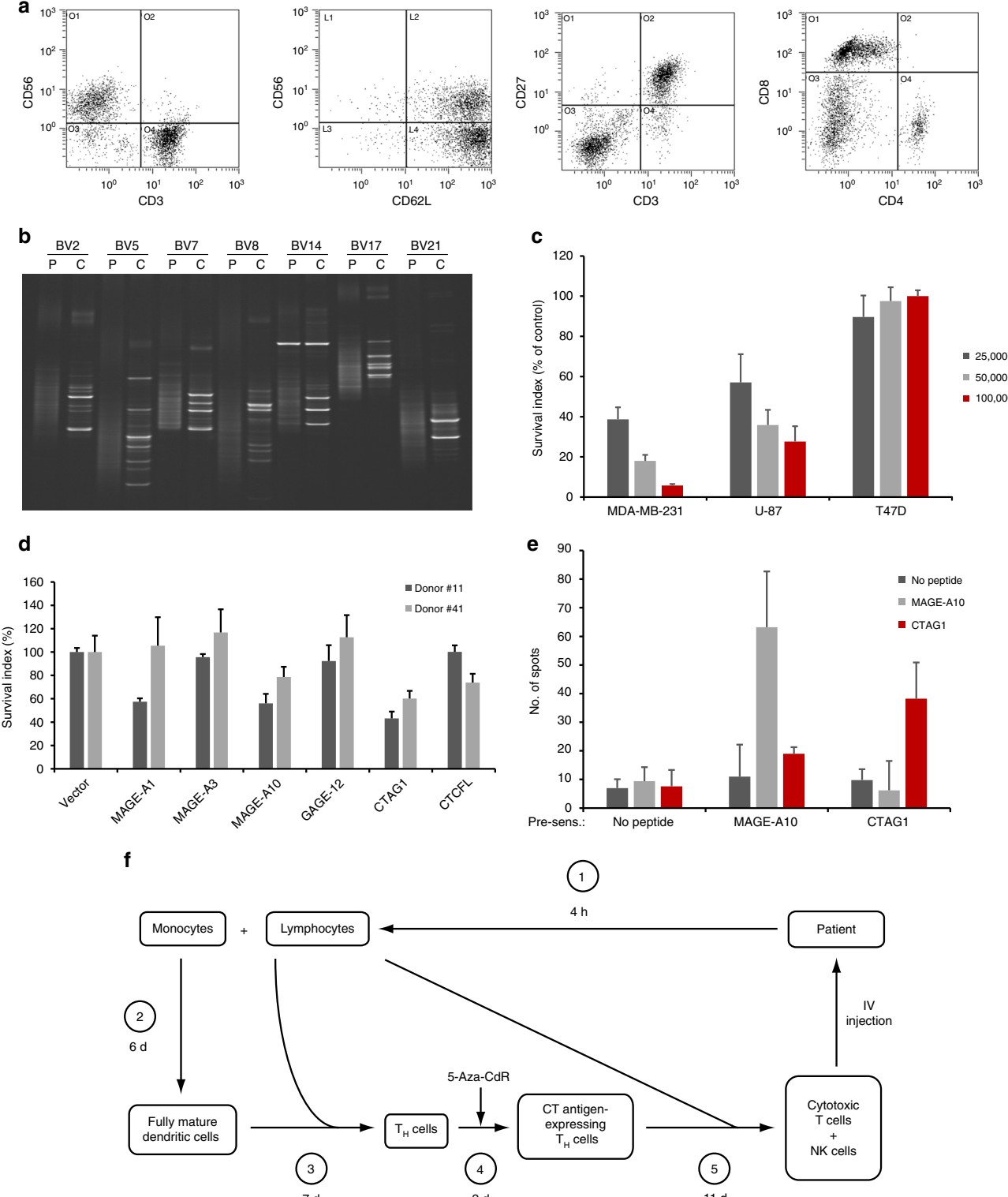

**Fig. 3** Generation of tumor-reactive lymphocytes by incubation of PBLs with DNA-demethylated $T_H$ cells. **a** Flow-cytometric analysis of lymphocytes generated by incubation of PBLs from a healthy donor with autologous DC-induced, 5-aza-cdR-treated lymphocytes for 11 days. **b** TCR clonotype mapping for comparative analysis of clonotypic T cells in untreated PBLs (P) and CTLs (C) from the same donor. **c** Dose-dependent killing of breast cancer and glioblastoma cells. MDA-MB-231 breast cancer (HLA-A2$^+$), U-87 glioblastoma (HLA-A2$^+$), and T47D breast cancer (HLA-A2$^-$) cells were incubated for 24 h with different numbers of cytotoxic lymphocytes prepared from a HLA-A2$^+$ donor and quantitified using the SRB assay. Data of triplicates are represented as means ± s.d. **d** Lysis of MEL-ST cells expressing single CT antigens by CD56-depleted cytotoxic lymphocytes. Representative results are shown for two of five healthy donors tested. Data of triplicates are represented as means ± s.d. **e** ELISPOT analysis. Cells were presensitized with peptides and dendritic cells for 7 days, as indicated. For ELISPOT assays, $5 \times 10^4$ cells per well were seeded in the presence or absence of peptides, and analysis was performed with five replicates. **f** Diagram showing the procedure for generation of cytotoxic lymphocytes used in the clinical trial

production of IFN-γ, suggesting that the CTLs recognize the antigen-presenting cells used for immunization. This effect could be partially blocked by an MHC class I-specific antibody (Supplementary Fig. 11). Residual anti-T cell activity in these cultures may be due to the generation of autoreactive CD4+ cells in response to 5-aza-CdR treatment, as previously reported[23].

We next incubated CTLs with human breast cancer cell lines and immortalized melanocytes (MEL-ST). Induction of cytokine release by CTLs after contact with cancer cells was observed in an HLA class I-restricted manner, whereas only minimal production of cytokines was seen after contact with immortilized melanocytes (Supplementary Fig. 12). Furthermore, the production of IFN-γ could be blocked by an HLA class I-specific antibody (Supplementary Fig. 13). Dose-dependent elimination of breast cancer and glioblastoma cells by CTLs was demonstrated using a sulforhodamine-based assay (Fig. 3c) and by real-time cell analysis (Supplementary Fig. 14). Of note, generation of tumor-reactive T cells was dependent on incubation with 5-aza-Cdr-treated cells; no tumor reactivity was observed in cultures grown in the absence of any stimulator cells (Supplementary Fig. 15). Although 5-aza-Cdr-induced CT antigen expression in both CD4+ and CD8+ cells (Supplementary Fig. 5), CD4+ cells were the most efficient stimulatory cells in terms of the number of lymphocytes generated (Supplementary Table 3). To further investigate tumor specificity, we incubated cells from a human breast cancer biopsy with cytotoxic lymphocytes prepared from the same patient's blood. As shown in Supplementary Fig. 16, cytotoxicity was induced in the tumor cells, but not in the normal cells in the same culture.

**Target recognition**. To evaluate the antigen specificity of CTLs generated by exposure to 5-aza-CdR-treated $T_H$ cells, we developed a functional assay using MEL-ST cells transduced with each of six CT antigens (MAGE-A1, MAGE-A3, MAGE-A10, CTCFL, GAGE-12, and CTAG1). Lytic activity against cells expressing single antigens was demonstrated; however, the number and profile of antigens recognized varied among donors, with MAGE-A10 and CTAG1 being the most commonly recognized targets (Fig. 3d, Supplementary Table 4). The specificity of the immune reaction was confirmed by ELISPOT analysis using immunodominant HLA-A2-restricted MAGE-A10 and CTAG1 epitopes (Fig. 3e).

**A phase 1 study in late-stage GBM**. Adoptive transfer of cytotoxic lymphocytes was next tested in a phase 1 trial of late-stage GBM. Between September 2011 and November 2012, a total of 25 patients with radiologically confirmed recurrent GBM were enrolled into the study. The baseline characteristics of these patients are presented in Supplementary Table 5, including information on first and second-line treatments. Apart from prednisolone, the patients received no other treatment for their GBM during the present trial. Data on immunohistochemical analysis of expression of selected CT antigens in diagnostic tumor biopsies are shown in Supplementary Fig. 17 and Supplementary Table 6. The protocol for generating cytotoxic lymphocytes used in the clinical trial is summarized in Fig. 3f. Induction of MAGE-antigen expression by 5-aza-CdR in $T_H$ cells from these patients, used as a marker of successful DNA demethylation, is shown in Supplementary Table 7. We did not detect expression of any of these CT antigens in untreated lymphocytes.

The trial had two parts. The first comprised three rounds of treatment with cytotoxic lymphocytes (freshly-prepared from repeated blood draws; median, $6.8 \times 10^7$ cells per treatment; Supplementary Table 8) at 4–5 week intervals without prior lymphodepletion, and evaluation after 20 weeks. In the second

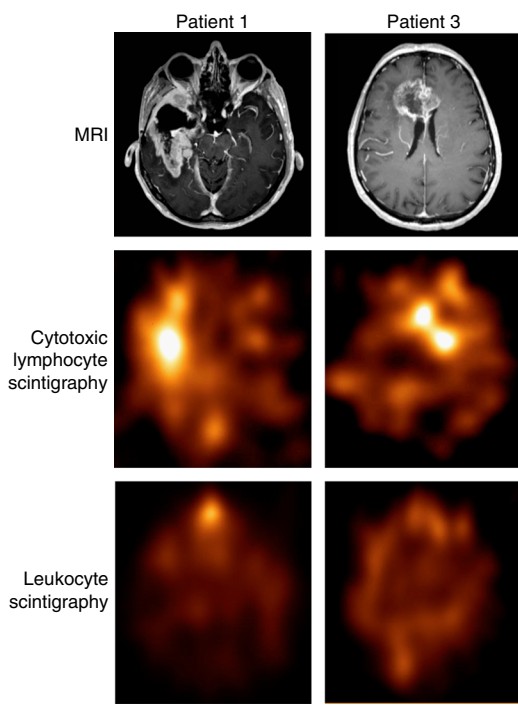

**Fig. 4** Homing of cytotoxic lymphocytes to the tumor site. The brain tumors are depicted on the MRI (upper panel). Cytotoxic lymphocyte scintigraphy (second panel) shows true positive foci with 3.1 and 1.9 times higher uptake of radiolabeled cells in the brain tumor than in the contralateral normal background area of Patients 1 and 3, respectively. Baseline leukocyte scintigraphy is normal with no focal accumulation of radiolabeled cells in the tumor of either patient (lower panel). In Patient 1, discrete leukocyte accumulation is seen in the nose compatible with slight inflammation

part, patients compliant after the 20-week study period received continued treatment with ad hoc injections of cytotoxic lymphocytes.

Homing to tumor: We first used scintigraphy to investigate the ability of therapeutic cells to home to the tumor site. Cells (median $30 \times 10^6$; range $1–100 \times 10^6$) were labeled with $^{111}$In-tropolone, and radioactive accumulation in the brain was measured 24 h after injection using single-photon emission computed tomography/computed tomography (SPECT/CT) imaging. Scintigraphic examination was performed in 15 patients during the first round of treatment, with repeat examination in six of these patients during the second round of treatment. Radioactive foci in the brain were observed in 14 of the 15 patients and in 20 out of the 21 individual examinations. Of 23 radioactive foci detected, 20 were located at the tumor site (Fig. 4). As a control, we performed scintigraphic examination in six patients before the first round of treatment using $^{111}$In-tropolone-labeled, non-activated leukocytes (median $4.0 \times 10^8$ cells; range $1.7–6.0 \times 10^8$ cells). One patient showed focal accumulation in the brain corresponding to part of the tumor. No radioactive uptake in the brain was observed in the five other patients receiving non-activated leukocytes.

Safety: During the 4-week period from the first blood donation to the first intravenous injection of therapeutic cells, two patients were withdrawn from the study because of disease progression. Among the 23 patients who received at least one round of treatment, 32 adverse events were reported in 19 patients (Supplementary Table 9), and 23 serious adverse events were reported in 17 patients (Supplementary Table 10). All events were considered the result of tumor progression and unrelated to

treatment. Nevertheless, because of the localization of the tumor in the brain, it cannot be excluded that some neurological symptoms (e.g., nausea, epilepsy, aphasia and transient paralysis) might be caused or aggravated by treatment-induced inflammatory responses.

Clinical activity: The flow of the study is shown in Supplementary Fig. 18. In total 10 of the 25 patients received all three scheduled rounds of treatment and were alive at the 20-week evaluation. The remaining 15 patients died from continued progressive disease before ($n = 14$) or immediately after ($n = 1$) the 20-week evaluation, having received no ($n = 2$), one ($n = 9$) or two ($n = 4$) rounds of treatment. Three patients who received three rounds of treatment could not be evaluated with respect to antitumor response after 20 weeks; one because of severe compliance problems due to a previously known respiratory insufficiency (Patient 14), and two because of non-compliance due to tumor progression (Patients 9 and 24).

Treatment-induced changes in the numbers of peripheral blood leukocytes were evaluated using data obtained immediately before and 1–2 days after injection of cytotoxic lymphocytes. Overall, each of the three injections of cytotoxic lymphocytes was accompanied by a transient increase in lymphocyte counts and a transient decrease in neutrophil counts, whereas there were no significant changes in total leukocyte counts (Supplementary Fig. 19).

During the 20-week study period (130–150 days from baseline), three patients showed decreases in tumor size of, respectively, 18% (at 63 days; Patient 14), 28% (at 131 days; Patient 6), and 31% (at 142 days; Patient 3) as assessed by MRI. Two patients initially showed increases in tumor size of 138% (at 36 days; Patient 5) and 238% (at 121 days; Patient 20; Supplementary Fig. 20), respectively, and stabilization thereafter (Fig. 5).

Under approved compassionate use, seven patients who were compliant after the 20-week study period received 1–18 additional lymphocyte injections until death or non-compliance

(Supplementary Table 11). MRI measurements were obtained in connection with the in-hospital treatment with therapeutic cells. One patient (Patient 5) showed stable disease on MRI but died 171 days after the first injection, one month after an ultrasound investigation had identified multiple metastases to the liver, consistent with adenocarcinoma (no biopsy was performed due to the poor clinical condition of the patient). Three patients (Patients 2, 18 and 24) showed tumor progression on MRI and underwent further surgical tumor debulking but died due to disease progression, 440, 267 and 164 days after the first lymphocyte injection, respectively.

The three remaining patients (3, 6, and 20) showed decreases in tumor size of, respectively, 86% (at day 668), 63% (at day 418), and 100% (at day 549) as assessed by MRI (Fig. 5). The corresponding assessments by FET-PET showed marked reductions in metabolically active tumor volumes in Patients 3 (Fig. 6) and 6, from 65 to 32.8 cm$^3$ and from 15.4 to 2.8 cm$^3$, respectively. In patient 20, the metabolically active tumor volume was 24 cm$^3$ at enrollment, with an increase to 43 cm$^3$ at day 261 and then a decrease to 29.5 cm$^3$ at day 791 (Supplementary Fig. 20). These three patients all experienced improvements in cognitive and motor functions. It is noteworthy that one of these patients (Patient 3) had no increase in tumor size during a period of nearly nine months between two injections of cytotoxic lymphocytes. There were no adverse events observed in any of the seven patients receiving continued treatment.

Patient 3 died 717 days after initiation of treatment from a small metastatic lesion in the brainstem (visualized on MRI; autopsy could not be performed). Patient 6 died from an

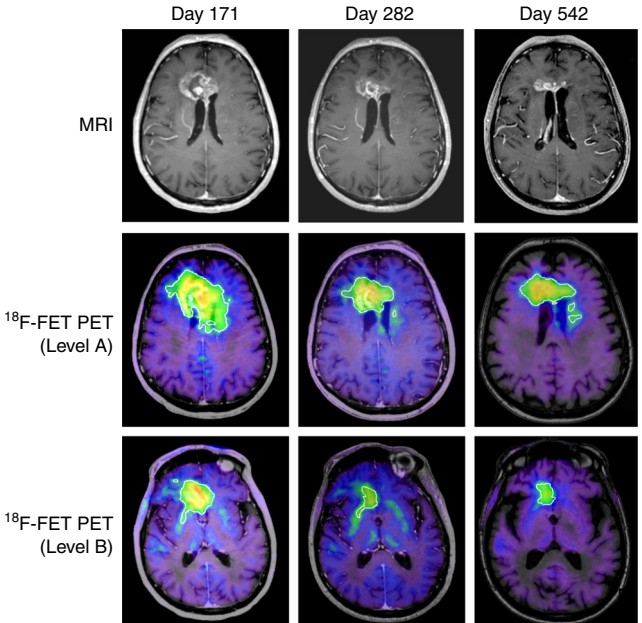

Day 171          Day 282          Day 542

**Fig. 6** Treatment response in Patient 3. This 76-year-old patient had a bilateral GBM tumor and was treated with radiation and concomitant temozolomide, with tumor progression four months into the temozolomide regimen. The patient received a total of eight injections of autologous cytotoxic lymphocytes, and stable disease was achieved within one month after the first injection. The MR images (top panel) illustrate a continuous reduction in size of the contrast-enhancing lesion by 63% over a course of 542 days. The middle and lower panels represent FET-PET images at two different levels of the brain, showing a 50% reduction in metabolically active tumor volume over the course of one year. The green areas represent the invasive parts of the tumor

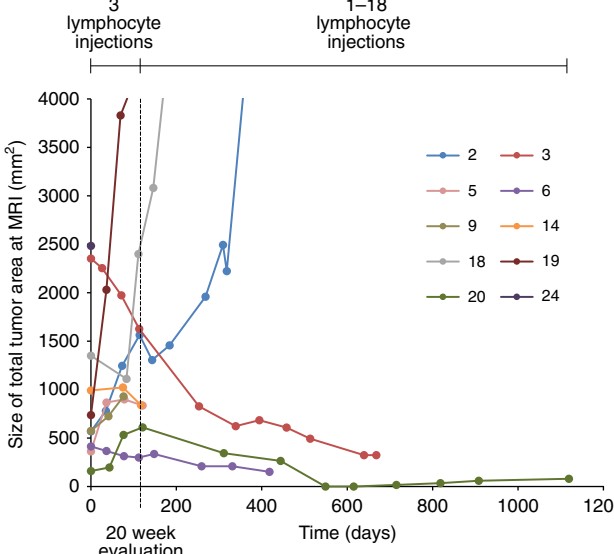

**Fig. 5** Treatment responses in 10 patients with relapsed GBM. All patients shown had at least three injections of cytotoxic lymphocytes. Shown is the size of the contrast-enhancing tumor on MRI. In general, the MRI measurements were obtained in connection with the in-hospital treatment with therapeutic cells

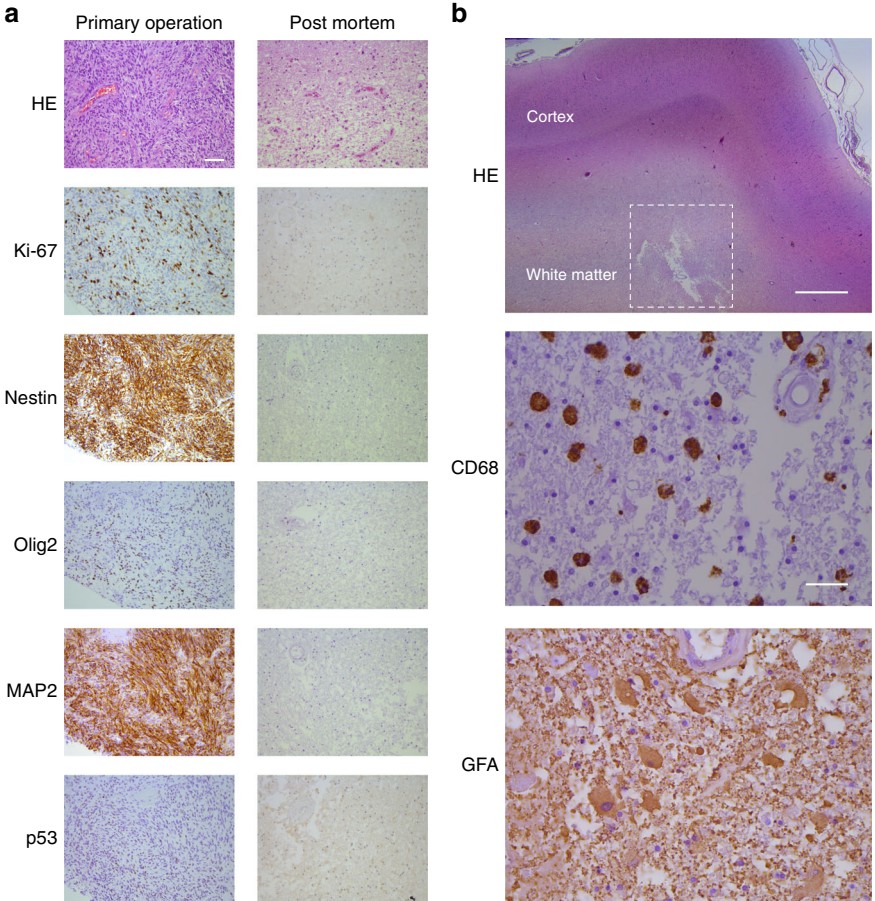

**Fig. 7** Treatment response in Patient 6. **a** Immunohistochemical analysis of Ki-67, Nestin and GBM markers in biopsies obtained at diagnosis (tumor) and autopsy (previous tumor location), ×200 objective; scale bar = 50 μm. **b** Low-magnification (×12.5 objective; scale bar = 1 mm) overview of the affected brain region from the autopsy specimen (HE) and immunohistochemical analysis of the macrophage lineage marker CD68 and the astrocyte marker GFA, ×400 objective; scale bar = 25 μm. The previous tumor location area analyzed by immunohistochemistry, including the necrotic cavity, is indicated by dashed lines

unrelated heart condition 545 days after initiation of treatment. Post-mortem histopathological examination of the brain from this patient using a panel of immunohistochemical markers (the proliferation marker Ki-67, the stem-cell marker Nestin and the tumor markers Olig2, MAP2, and p53) revealed complete absence of cancer cells at the previous tumor site, in the surrounding brain parenchyma and in the contralateral hemisphere. Staining for CD68 and GFA demonstrated the presence of cells of the macrophage lineage and gliosis with hypertrophic glial cells (Fig. 7). Staining for CD8 revealed rare scattered T cells with occasional clusters around a necrotic cavity at the previous tumor location (Supplementary Fig. 21). Patient 20 experienced relapse and died 1384 days after initiation of treatment (visualized on MRI; autopsy could not be performed).

## Discussion

We report a novel approach to cancer immunotherapy based on the adoptive transfer of cytotoxic lymphocytes with specificity for antigens induced by DNA demethylation. The biological rationale for this therapeutic strategy was to target the widest possible repertoire of CT antigens, as numerous studies reported great variation in expression of individual CT antigens across tumor types and stages, as well as within tumors[7]. We posited that induction of an immune response against multiple CT antigens

could increase therapeutic efficacy and decrease the risk of resistance development relative to single-target therapies. The technical rationale was to induce expression of endogenous CT antigens in autologous antigen-presenting $T_H$ cells by pharmacological demethylation. The key discovery that made this approach feasible was the ability to utilize autologous fully mature DCs to induce non-specific lymphocyte proliferation ex vivo and enrich for $T_H$ cells. After induction of CT-antigen expression by DNA demethylation, these $T_H$ cells could readily be used as antigen-presenting cells.

Exposure of PBLs to DNA-demethylated $T_H$ cells led to the generation of tumor-reactive CTLs and NK cells. Since DNA demethylation can, in theory, induce expression of hundreds of CT antigens[7], CTLs targeting a specific CT antigen would be expected to represent only a small fraction of the total CTL population. Still, specific lysis was induced not only in tumor cells expressing multiple CT antigens, but also in primary cells expressing single CT antigens. It is noteworthy that the T cell precursor frequencies for two of the recognized antigens in our testing panel, MAGE-A10 and CTAG1, were reported to be significantly higher in healthy donors compared to other investigated CT antigens[24, 25]. It is well known that one of the leading factors for the induction of an immune response against a specific antigen is the T cell precursor frequency[26]. Hence, our results provide proof-of-principle that it is possible to generate a polyvalent response to a variety of CT antigens, provided that CTL

precursors are present at certain levels. Lack of reproducible detection of immune responses against other tested CT antigens probably reflects a low frequency of their respective precursors, and does not exclude generation of a CTL response against these antigens, albeit at low levels. One might expect that, after multiple rounds of treatment with cytotoxic lymphocytes in the same patient, the proportions of CTLs originally present at low frequencies would increase and hence broaden and strengthen the immune response against tumor cells.

Current protocols for ex vivo expansion of therapeutic T cells tend to drive these cells into a state of high differentiation and replicative exhaustion[27]. Studies in various animal models have shown that, despite their ability to effectively kill tumor cells, highly differentiated effector memory T cells and terminally differentiated effector T cells have low antitumor activity compared with less differentiated central memory T cells[28, 29]. This loss of therapeutic potential may be attributed at least partially to the loss of expression of CD27, a molecule required for long-term T-cell survival[30, 31], and CD62L, a cell adhesion molecule that allows lymphocytes to enter secondary lymphoid tissues by transendothelial migration via high endothelial venules and hence is critical for the generation of long-lived pools of memory cells[32]. Several lines of evidence suggest that CD62L expression may also be a major determinant of the ability of lymphocytes to infiltrate tumors[28, 33]. The two major cell populations generated by the immunization protocol described herein, CD8+ T cells and NK cells, both expressed CD62L, consistent with an early differentiation state[34, 35], and with the potential to infiltrate tumors. Indeed, in our study, scintigraphic analysis demonstrated homing of the injected cytotoxic lymphocytes to the tumor site. Whether the CD8+ T component, by virtue of its phenotype (including the expression of CD27), is capable of inducing a long-lasting immune response, thus obviating the need for repeated injections, is a topic for future studies.

The median survival after diagnosis of GBM is 14.6 months, making it one of the most lethal forms of cancer and one of the most challenging to treat[10, 11]. In the present study, disease control was achieved in five of ten patients receiving the three scheduled injections of therapeutic cells, with long-term tumor regression in three of these patients who were all in the terminal phase of their disease with rapidly growing tumors. The lack of objective responses in the majority of our patients should be interpreted in the context of their poor clinical condition, with a life expectancy of <6 months. Similar to several other forms of immunotherapy[36], a disadvantage of our approach may be the slow kinetics of the induced antitumor response. Indeed, in three of the responding patients, disease stabilization became manifest several months after the first injection of cytotoxic lymphocytes, with initial increases in tumor burden. Hence, in many patients with relapsed GBM and rapid progression, the disease may become fatal before an efficient antitumor response can be generated. Alternatively, the lack of response may have been the result of tumor-mediated immune evasion mechanisms[37] or the lack of CT-antigen expression by tumor cells.

The extent and type of side effects associated with cell-based immunotherapeutic cancer therapies depend on the targeted antigens. For example, depigmentation due to autoimmune melanocyte destruction is a common side effect of therapies targeting melanocyte differentiation antigens such as MART-1 and gp100[38], and targeting of CD19 (a B cell-specific antigen) in leukemias results in conditions associated with loss of normal B cells, including lymphopenia and degammaglobulinemia[5, 39]. Of particular relevance to this study, severe treatment-induced cardiovascular and neurological toxicity has been observed in patients treated with autologous T cells genetically engineered to express a TCR targeting the widely-expressed CT antigen,

MAGE-A3[4, 40]. These adverse effects were likely due to off-target reactivity of the genetically engineered TCRs used in these studies, which were affinity matured to achieve an avidity in excess of the natural TCR. Our approach did not involve any modification of TCRs, which may explain the absence of toxic, allergic or autoimmune reactions despite theoretically targeting a broad spectrum of CT antigens. Nevertheless, it cannot be excluded that some of the neurological symptoms observed in our GBM trial could be caused by treatment-induced inflammatory responses in the brain, elicited by on-target on-tumor effects of the therapeutic cells.

In summary, we show that adoptive transfer of lymphocytes with reactivity to antigens induced by DNA demethylation is a potential approach to anticancer immunotherapy. This minimally invasive strategy may overcome important limitations of current immunotherapies in terms of antigen coverage and differentiation state of therapeutic lymphocytes, hence providing a general strategy for treatment of a wide range of human cancers, potentially in combination with checkpoint inhibitors to remove T-cell inhibition. For tumors with low expression of CT antigens, immunogenicity may be enhanced by systemic treatment with DNA-demethylating drugs to selectively upregulate CT-antigen expression in the tumor cells prior to immunotherapy[41, 42]. Provided that the induced immune response is persistent and has no long-term adverse effects in healthy persons, the present approach may also open new possibilities for cancer prevention.

## Patients and methods

**Patients**. The study was performed at the department of Neurosurgery, Rigshospitalet, Copenhagen, Denmark. Inclusion criteria were radiologically confirmed relapsed GBM on magnetic resonance imaging (MRI) and a Karnofsky performance score of >50% in patients >18 years of age. Exclusion criteria were blood hemoglobin <6 mM, positive test for HIV, hepatitis B and C and syphilis. Informed consent was obtained from all patients. The study was conducted in accordance with the Declaration of Helsinki and approved by The Committee on Biomedical Research Ethics of the Capital Region of Denmark (H-1-2011-068).

**Study design and treatment**. In this open, prospective phase 1 study, patients with relapsed late-stage GBM received a total of three intravenous injections (injection site: peripheral vein of the arm or hand) of cytotoxic lymphocytes every 4–5 weeks and were monitored during a 1–2 day hospitalization period. For each patient, successive batches of cytotoxic lymphocytes were prepared from freshly drawn blood. Immediately prior to the first injection, the prednisolone dose was set at a maximum of 37.5 mg/day; however, any subsequent symptoms related to increased tumor edema were treated ad hoc with higher doses. During the trial with cytotoxic lymphocytes, the patients received no other treatment for their GBM than prednisolone. All patients included in the study were evaluated for safety (primary endpoint). Patients who were compliant 6 weeks following the first injection of cytotoxic lymphocytes ($n = 14$) were evaluated for safety and tumor control (secondary endpoint). The protocol had an overall duration of 20 weeks from the initial blood donation to the last visit. The actual treatment started in week 5 with the first injection of cytotoxic lymphocytes. After the 20-week study period, patients who were compliant were given 1–18 additional injections of cytotoxic lymphocytes on a compassionate basis.

Total numbers of leukocytes, lymphocytes and neutrophils in patient blood were measured immediately before and 1-2 days after injection of cytotoxic lymphocytes. Disease assessment was performed by MRI and O-(2-[18F] fluoroethyl)-L-tyrosine positron emission tomography (FET-PET) (Supplementary Methods). Homing of radiolabeled cells was analyzed by scintigraphy (Supplementary Methods). Safety and tolerability assessments are described in Supplementary Methods. The present clinical trial is listed on-line at https://eudract.ema.europa.eu as protocol EudraCT Number 2011-002180-22, and at http://www.clinicaltrials.gov with identifier NCT01588769, and was monitored according to Good Clinical Practice by an independent unit in Copenhagen.

**Cell lines**. Breast cancer cell lines MDA-MB-231, T47D, CAMA-1 and MCF-7 and the glioblastoma cell line U-87 were purchased from ATCC (Manassas, VA). FM3 and FM3.29 melanoma cell lines have been previously described[43]. All cell lines were tested for mycoplasma contamination. Cells were cultured in RPMI 1640 medium supplemented with 10% FCS. Data on CT-antigen expression in these cell lines are provided in Supplementary Table 12.

**Preparation of cytotoxic lymphocytes**. The standard 26-day procedure for preparation of cytotoxic lymphocytes is outlined in Fig. 3f and consists of the following steps:

Preparation of mature dendritic cells (DCs): Preparation of mature dendritic cells was essentially as described[44]. Mononuclear cells were isolated as described[44]. After last washing, cells were re-suspended in cold DPBS and counted using the model Z2 Coulter Counter. Monocyte concentration was determined by gating the corresponding cell peaks. Generation of DCs was performed in T225 tissue culture flasks pre-treated with 30 ml of 5% human AB serum in RPMI 1640. After removal of pre-treatment medium, 40 ml of a cell suspension containing $5 \times 10^7$ monocytes in AIM-V medium were added. After 30 min of incubation at 37 °C, non-adherent lymphocytes were collected, adherent monocytes rinsed twice with pre-warmed RPMI 1640 medium and further cultured in AIM-V medium. The collected lymphocytes were frozen in several aliquots. After overnight incubation (day 1), and at day 3, GM-CSF and IL-4 (both from Gentaur, Belgium, or CellGenix, Germany) were added to the cells at final concentrations of 100 ng/ml and 25 ng/ml, respectively. At day 4, IL-1β, IL-6, TNF-α (all from Gentaur) and PGE2 (Sigma) were added at final concentrations of 10 ng/ml, 1000 IU/ml, 10 ng/ml and 0.2 μg/ml, respectively. At day 6, non-adherent cells were harvested, counted and used for our experiment, and the remainder were frozen in aliquots of $2–3 \times 10^6$ cells.

Induction of CD4$^+$ cell proliferation and 5-aza-CdR treatment: A mixture of thawed non-adherent lymphocytes and DCs (in 10:1 ratio) was incubated in AIM-V medium containing 1% autologous serum. At day 1, IL-2 was added at final concentration of 25 IU/ml. Fresh medium and IL-2 were added at days 4 and 6. At day 7, cells were collected, counted, washed, resuspended in fresh medium, and, after addition of 150 IU/ml of IL-2 and 10 μM 5-aza-CdR, cultured for 2 days.

Induction of cytotoxic lymphocytes: A mixture of 5-aza-CdR-treated cells and thawed non-adherent lymphocytes ($0.5 \times 10^6$/ml each) was incubated in AIM-V medium containing 2% autologous serum. After 2 days of incubation, IL-2 was added at a final concentration of 25 IU/ml. Fresh medium and IL-2 were added at days 5, 7, and 9. At day 11, a portion of the cells was used for flow cytometry and, after release of the product, cells were prepared for injection by washing and placing into a syringe in 21 ml of Plasma-Lyte (Baxter International, Deerfield, IL) with the addition of 5% autologous serum.

**Flow cytometry**. For determination of surface expression of different markers on the cells, we used the directly conjugated antibodies listed in Supplementary Table 13. The recommended isotypic controls were used for the phenotyping of the cells. The cell samples were analyzed using FC500 MPL Flow Cytometer (Beckman Coulter) and the CXP analytical software (Beckman Coulter).

**Cytokine production**. CD56-depleted cells were isolated from cytotoxic lymphocytes prepared from the blood of HLA-A2$^+$ donors using CD56 Microbeads (Miltenyi Biotec) according to the manufacturer's instructions. After isolation, lymphocytes were washed once, re-suspended at $5 \times 10^5$/ml, and added to a panel of breast cancer cell lines. Tumor cells ($2 \times 10^4$) were seeded in a 48-well plate in 0.5 ml of RPMI-1640 medium with addition of 10% FCS one day before the test. 0.5 ml of the suspension of isolated lymphocytes containing $2.5 \times 10^5$ cells was added to the tumor cells. Cells were cultured together for 18–20 h. Production of IFN-γ and TNF-α in the culture supernatant was measured by ELISA (see below).

**Sulforhodamine B assay**. Cell monolayers were incubated with cytotoxic lymphocytes for 24 h. The lymphocytes were then removed from the wells by gravity sedimentation. Briefly, the wells were filled with DPBS, sealed and turned upside-down for 20 min, after which the sealer was detached without changing the plate's position, and excess liquid removed by gentle blotting on a paper towel. Adherent cells were fixed, stained with 0.06% Sulforhodamine B (SRB) and processed according to a previously described version[45] of the SRB assay[46]. The dye was extracted and absorbance measured at 550 nm for cultures incubated with or without lymphocytes. The survival index was determined as the ratio between these cultures, with the control culture (without lymphocytes) set as 100%.

**Cytokine ELISA**. Concentrations of IL-12p70, IFN-γ, and TNF-α in cell culture supernatants were measured by sandwich ELISA using appropriate "Ready-Set-Go" kits (eBioscience). The procedure was essentially as recommended by the manufacturer, except that when measuring IL-12p70, the standards and samples in triplicates were incubated with detection antibodies at room temperature for 2 h followed by overnight incubation at 4 °C.

**Real-time monitoring of cell viability**. Cytolytic activity of lymphocytes was monitored in real-time cytotoxicity assays using the iCELLigence system (ACEA Biosciences, San Diego, CA, USA). Tumor cells were seeded at a density of $3 \times 10^4$ cells per well in a total volume of 400 μL of RPMI 1640 medium with 10% FCS. After 20-24 h initial incubation, 100 μl of culture medium were removed, and $0.2 \times 10^6$ lymphocytes were added in 200 μl of AIM-V medium. Killing of tumor cells was associated with a decrease in cell impedance (measured as Cell Index), and was monitored every 15 min for 20–25 h and presented as normalized Cell Index.

**Immunohistochemistry**. Surgical specimens were routinely formalin-fixed, paraffin-embedded and sectioned at 4 μm thickness. Immunohistochemical staining was performed on a BenchMark ULTRA IHC Staining module using OptiView DAB/Ventana, Roche, according to OptiView DAB IHC v5 protocol. Sections were first deparaffinized. Antigen retrieval was achieved by incubating the sections at 100 °C, pH 8.5 for 32 min. The sections were incubated in hydrogen peroxide for peroxidase inhibition, followed by incubation with the primary antibody. The antibodies used are listed in Supplementary Table 13.

**Transduction of immortalized melanocytes**. Immortalized human melanocytes, MEL-ST ([47]; a gift from R.A. Weinberg, Whitehead Institute for Biomedical Research and Massachusetts Institute of Technology, Cambridge, Massachusetts, USA), were transduced with lentivirus to express CT antigens. Six CT-antigen open reading frames (MAGE-A1, MAGE-A3, MAGE-A10, CTCFL, GAGE12, and CTAG1) were cloned from testes RNA into a lentiviral expression vector (CD513B-1, System Biosciences). As controls, we used empty vector. The vectors were packed in HEK293T cells into lentiviral particles, which were used to transduce the immortalized melanocytes. Four days after transduction, the cells were selected with 0.25 μg/ml puromycin for one week. The vector contains the gene encoding the green fluorescent protein (GFP), and successful transduction was confirmed in a fluorescence microscope or with flow cytometry. The cells were grown in DMEM medium supplemented with 5% FBS.

**ELISPOT analysis**. Autologous mature dendritic cells were seeded into 24-well plates ($10^5$ cells per well) in 1 ml of AIM-V medium supplemented with 2% of autologous serum. HLA-A2-restricted peptides from CTAG1 (p157-165, SLLMWITQC, IBA, Goettingen, Germany) and MAGEA10 (p254-262, GLYDG-MEHL, KJ Ross-Petersen ApS, Copenhagen, Denmark) were added at a concentration of 20 μg/ml, and after 30 min incubation at room temperature, autologous CD56-negative cells were added. After 48 h, IL-2 (50 IU/ml) was added to the culture wells and cells were cultured for additional 5 days. IFN-γ ELISPOT assays were performed in 96-well precoated plates using the 3420-4APW-2 Human IFN-γ ELISpot$^{PLUS}$ Kit (ALP) (MABTECH, Stockholm, Sweden). After washing with PBS, wells were blocked with 10% FCS for 30 min. Fifty thousand CD56-negative cells, either presensitized with peptide for 7 days (see above) or not presensitized (cultured under the same conditions without the addition of peptides) were added to each well in the presence or absence of peptides. After overnight incubation, plates were processed according the manufacturer's procedure. Counting of spots was done automatically using the ImmunoSpot Reader (Cellular Technologies Limited) and the ImmunoSpot 4.0 software.

**RT-PCR**. Total RNA was isolated from tumor cells and lymphocytes according to protocol 1[48], reconstituted in RNase-free water, treated with DNase from the Turbo-DNA-free$^{TM}$ kit (Invitrogen), and reverse transcribed using SuperScript III reverse transcriptase using random hexamers and oligo(dT$_{24}$) as primers. cDNA and DNase-treated RNA (negative control) were amplified using a block cycler and gene-specific primers (Supplementary Table 14) for 32-35 (MAGEs) and 37-43 (GAGE$_{3-7}$, CTAG1 and CTCFL) cycles under previously reported conditions[49, 50]. For qPCR, total RNA was isolated using the NucleoSpin RNA Plus kit (Macherey-Nagel), and reverse transcribed using SuperScript$^{TM}$ IV VILO$^{TM}$ Master Mix with ezDNase$^{TM}$ Enzyme (Invitrogen) and random hexamers and oligo(dT$_{24}$) primers. cDNA was amplified in a LightCycler Nano (Roche) using the FastStart Essential DNA Green Master mix (Roche) and previously described primers and conditions[51], except that validated primer sets for MAGEA4, MAGEA10 and CTAG1B were VHPS-5479, VHPS-5478 and [5'-TCTCCATCAGCTCCTGTCTC-3' and 5'-CAAACATGTAAGCCGTCCTC-3'], respectively (Biomol). The data were normalized to GAPDH expression.

**TCR clonotype analysis**. Clonotypic T cells were detected by RT-PCR/DGGE-based TCR clonotype mapping, as previously described[22]. In brief, cDNA was amplified using primers for TCRBV regions 1–24. For BV regions that were expressed in both untreated PBLs and CTLs from the same donor, amplicons were analyzed in a 6% polyacrylamide, 20–80% denaturant gel (100% denaturant = 7 M urea and 40% formamide). The gel was run at 160 V for 4.5 h in TAE buffer (0.04 M Tris-acetate, 0.001 M EDTA) kept at a constant temperature of 56 °C, stained with ethidium bromide, and photographed under UV transillumination.

**DNA methylation analysis**. DNA was isolated from cultured cells and treated with sodium bisulfite using the EZ DNA Methylation-Gold$^{TM}$ Kit (Zymo Research). The methylation status of gene promoter regions was determined by methylation-specific melting curve analysis (MS-MCA)[52] using the LightCycler 2.0 (Roche) and the FastStart DNA Master SYBR Green I Kit (Roche), and by pyrosequencing on a PyroMark Q24 platform (Qiagen). Primer sequences are listed in Supplementary Tables 15 and 16.

**Cell-based ELISA**. The procedure was essentially as described for quantification of gp100[53] except that ELISA optical density readings obtained for two-fold serial

dilutions of cells incubated with supernatants of the MAGE-specific 57B clone[21] were corrected for unspecific binding of isotype-matched IgG1 at the same concentration. The optical density was measured at 630 nm for the non-terminated reactions and at 450 nm after termination with HCl.

**Immunoblotting**. The total protein concentration was measured using the Pierce BCA Protein Assay Kit (Thermo Scientific). Aliquots were lysed in Laemmli buffer, boiled for 5 min and separated on 12% NuPAGE Bis-Tris gels in MOPS buffer (Invitrogen). Proteins were transferred to a nitrocellulose membrane (Hybond ECL, Amersham), blocked with 5% non-fat dry milk and 5% FCS in TBS, incubated with the primary antibody (57B) overnight at 4 °C and then with HRP-conjugated rabbit anti-mouse immunoglobulins (1:1000, Dako; Cat.no. P0260) for 1 h at room temperature. Reactive bands were visualized by Enhanced Chemiluminescence (ECL) detection (Amersham).

**Data availability**. All relevant data are available within the article and Supplementary Files, or available from the authors upon request.

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

## Acknowledgements

We thank Dr. G.C. Spagnoli for the 57B antibody, Dr. R.A. Weinberg for MEL-ST cells, Dorte Dyregaard for leading GMP production of cytotoxic lymphocytes, Pia Breum and Kamilla Overbeck Jensen for patient care and data collection, Tea Kirkegaard Nielsen for help with establishing ELISPOT assays, Claus Wedel for initial support and M.K. Occhipinti for editorial assistance. This work was funded by CytoVac A/S and supported by the Aase and Ejnar Danielsen Foundation, the Lisa and Gudmund Jørgensen Foundation, Bjarne Saxhof's Foundation, the Danish Research Council, and the Danish Cancer Society.

## Author contributions

A.F.K., and K.N.D. conceptualized and developed the method of adoptive T cell therapy described here. A.F.K., K.N.D., P.G., and J.J.F. performed most of the experiments. R.S.A., C.D., M.F.G. and H.J.D. performed antigen specificity experiments. W.F., and M.R.J. organized the clinical trial. W.F. treated the patients. J.M., T.L., A.W., and I.L. performed the imaging. H.B., and L.M. performed histopathological examinations. C.L.-E. managed the clinical data. A.F.K., K.N.D., P.G., and W.F. wrote the manuscript, and all authors read and approved the final manuscript.

## Additional information

**Competing interests:** A.F.K. and K.N.D. are named inventors on patents related to the project. A.F.K., K.N.D., J.J.F., C.L.-E., M.R.J., and W.F. are current or former employees of CytoVac A/S. The remaining authors declare no financial interests.

