## [Peer Review File · Nature Communications]

Reviewers' comments:

Reviewer #1 (Remarks to the Author):

Kirkin et al. describe a novel approach for the in vitro generation of T lymphocyte populations potentially suitable for adoptive therapy treatment of several types of cancer.

The novelty of the approach lies in the ability to generate T cell populations with a broad reactivity against so-called cancer testis antigens. Although this reactivity inevitably will vary from patient to patient, a multi-targeted T cell population approach could well prove to be a highly useful addition to the current arsenal immunotherapies.

Currently, approaches to harnessing T cells against cancer fall into three main categories, all suffering from various limitations: 1) the broadest immune response is brought about by checkpoint inhibitors but their use and efficacy is limited by toxicity, as they activate T cells indiscriminately and by the fact that they only work in tumours with a high mutational burden; 2) administration of a patient's expanded T cells isolated from a tumour sample has shown impressive efficacy in melanoma but the approach is constrained by the practical difficulties of obtaining a suitable biopsy and by the difficulties of expanding the T cells; 3) antigen targeted approaches involving, for example, Ab or T cell receptor transduced T cells, mainly target one particular antigen and, as tumours are heterogeneous, they will only address part of a cancer cell population unless they can be made to bring in a broader immune response.

In this landscape of approaches, the work of Kirkin et al is most interesting as it offers a way of targeting multiple antigens without the associated normal cell reactivity and toxicity. The potential of the approach is demonstrated both by in vitro and clinical data from a trial in recurrent glioblastoma.

This is an important piece of work that will of high interest both in the field and in general to the readers of Nature Communications. The experiments are well performed and the manuscript concisely and well written. The in vitro analysis of the reactivity of the generated T cells to various antigens and, in particular, to many types of cancer cell lines or dissociated tumour cells of different origin, would ideally have been far more comprehensive; this would have been informative in assessing the future potential of the approach. However, the interest for this work will be considerable and I would therefore recommend that publication of the manuscript is not held up. I hope the work will be followed up later with a more detailed analysis of the CT antigen responses and their potential against various types of cancer cells.

Bent K Jakobsen

Reviewer #2 (Remarks to the Author):

This is an interesting paper describing the potential of 5-azacytine treated autoreactive CD4+ T-cells to express cancer/testis antigens and to present these antigens to autologous CD8+ T-cells thereby generating cytotoxic T-cells with the potential to produce cytokines of a Th1 type in response to tumor cells expressing these antigens and to lyse the tumor cells in an HLA-restricted manner. However, there are several aspects of the data presented that require clarification. The results of the phase I trial with these T-cells are also of high interest but, again, several points regarding how the patients were treated and evaluated need to be clarified for meaningful interpretation. Thereafter, the following questions need to be addressed:

1. The introduction posits that non-specific activation of T-cells with PHA, rather than DCs, generates CD4+T-cells that express proteins such as MICA and MICB which render them sensitive to NK cells. However, among the T-cells responding in an autologous MLC to DC there is also some expression of MICA/MICB (Suppl. Fig 1). Do the authors have data directly comparing DC

stimulated vs PHA-activated T-cells as to MICA/MICB expression, and are the DC stimulated T-cells truly less sensitive to NK cells?

2. The data regarding induction of CT antigens induced by 5-azacytine is of great interest. Do the authors have any data regarding the duration of this alteration in CT protein synthesis post 5-azacytine exposure?

3. The data presented in Figure 1 do demonstrate augmentation of the generation of CD4+ T-cells. However, what is not clear is whether the induction of CT proteins is differentially observed in CD4 vs. CD8 T-cells. Are there such data? Also, what were the yields of the T-cells after stimulation with the 5-aza treated Th1 cells.

4. The partial blocking of INF γ production with the W632 antibody shown in suppl. Figure 9 against the 5-azacytine treated CD4+ cells is of interest when compared to the full blocking of cytokine generation in response to breast cancer cells in Suppl Figure 11. Does the residual anti-T cell activity represent autoreaction CD4+ T-cells? Or other possibilities?

5. Several specifics regarding the phase I trial need to be specified:

a) Three injections of cells are given: What is the timing of these infusions; weekly, monthly, other? This needs clarification.

b) What is the timing of the assays measuring lymphocytes "after" infusion in Suppl. Figure 17?

c) Were the 5-aza Th1, enriched APC sensitized T-cells used for treatment all generated at the beginning, as one lot, or were T-cell batches separately generated for each infusion? The latter is suggested from the discussion on page 15, para 1. This is not specified in the text or supplements. If successive lots of T-cells were generated, did the T-cells exhibit the same attributes and specificities? Were the doses given the same?

d) Other than steroids, did the patients receive any other treatment for their GBMs during the trial?

6. It is also difficult to interpret the SPECT/CT results without specifying how many labelled PBL or 5-aza Th1 sensitized CTLs were administered. This needs to be specified.

7. The histology of the tumors from the treated patient6 shows macrophages, but no mention is made of T-cells. Were there T-cells in the residual tumor, or in the necrotic lesions?

8. Given that the authors have been able to characterize the antigenic specificities, in terms of CT antigens, of the T-cells generated, are there any data demonstrating an increase in the frequency of MAGE-10 specific T-cells in the blood of those patients who achieved disease stabilization?

9. The point regarding the genetically engineered T-cells expressing a high avidity MAGE-3 specific TCR that had severe off-target toxicities is a good one, but, in fairness the MAGEA3 TCRs used in that study were not naturally selected but affinity matured to have an avidity in excess of the natural TCR.

Response to Reviewers' comments:

Reviewer #1

General comments:

Kirkin et al. describe a novel approach for the in vitro generation of T lymphocyte populations potentially suitable for adoptive therapy treatment of several types of cancer.

The novelty of the approach lies in the ability to generate T cell populations with a broad reactivity against so-called cancer testis antigens. Although this reactivity inevitably will vary from patient to patient, a multi-targeted T cell population approach could well prove to be a highly useful addition to the current arsenal immunotherapies.

Currently, approaches to harnessing T cells against cancer fall into three main categories, all suffering from various limitations: 1) the broadest immune response is brought about by checkpoint inhibitors but their use and efficacy is limited by toxicity, as they activate T cells indiscriminately and by the fact that they only work in tumours with a high mutational burden; 2) administration of a patient's expanded T cells isolated from a tumour sample has shown impressive efficacy in melanoma but the approach is constrained by the practical difficulties of obtaining a suitable biopsy and by the difficulties of expanding the T cells; 3) antigen targeted approaches involving, for example, Ab or T cell receptor transduced T cells, mainly target one particular antigen and, as tumours are heterogeneous, they will only address part of a cancer cell population unless they can be made to bring in a broader immune response.

In this landscape of approaches, the work of Kirkin et al is most interesting as it offers a way of targeting multiple antigens without the associated normal cell reactivity and toxicity. The potential of the approach is demonstrated both by in vitro and clinical data from a trial in recurrent glioblastoma.

This is an important piece of work that will of high interest both in the field and in general to the readers of Nature Communications. The experiments are well performed and the manuscript concisely and well written. The in vitro analysis of the reactivity of the generated T cells to various antigens and, in particular, to many types of cancer cell lines or dissociated tumour cells of different origin, would ideally have been far more comprehensive; this would have been informative in assessing the future potential of the approach. However, the interest for this work will be considerable and I would therefore recommend that publication of the manuscript is not held up. I hope the work will be followed up later with a more detailed analysis of the CT antigen responses and their potential against various types of cancer cells.

Bent K Jakobsen

Response:

We appreciate that the limited panel of cell lines analyzed in this study represent only a few cancer types, and that a broader screening of cancer cell lines will provide important information about the versatility of our approach. Part of this work is currently under way in preparation for upcoming clinical trials of patients with other cancer types.

Reviewer #2:

General comments:

This is an interesting paper describing the potential of 5-azacytine treated autoreactive CD4+ T-cells to express cancer/testis antigens and to present these antigens to autologous CD8+ T-cells thereby generating cytotoxic T-cells with the potential to produce cytokines of a Th1 type in response to tumor cells expressing these antigens and to lyse the tumor cells in an HLA-restricted manner. However, there are several aspects of the data presented that require clarification. The results of the phase I trial with these T-cells are also of high interest but, again, several points regarding how the patients were treated and evaluated need to be clarified for meaningful interpretation.

Response:

We hope that the following responses and the additional experiments performed will satisfactorily address the points raised. In addition, we have deleted a sentence in the first paragraph of the Introduction that we felt was misplaced and potentially misleading given recent developments in the field.

Comment #1:

The introduction posits that non-specific activation of T-cells with PHA, rather than DCs, generates CD4+T-cells that express proteins such as MICA and MICB which render them sensitive to NK cells. However, among the T-cells responding in an autologous MLC to DC there is also some expression of MICA/MICB (Suppl. Fig 1). Do the authors have data directly comparing DC stimulated vs PHA-activated T-cells as to MICA/MICB expression, and are the DC stimulated T-cells truly less sensitive to NK cells?

Response:

The information about MICA/MICB expression in PHA-activated T-cells was from the literature, and we have now performed additional experiments to address this specific question. In these experiments, we did not observe any notable difference in the expression of MICA/MICB in DC-stimulated vs PHA-activated T cells. The ratio of CD4+/CD8+ cells, however, was significantly lower in PHA-activated cells, which may have implications for their ability to act as stimulatory cells (see below; Comment #3). The most important rationale for using an autologous cell-based method for activation of T cells was that employing a foreign antigen, such as PHA, would have made approval of the therapeutic product for clinical use difficult, at least in Denmark.

Based on these new findings, we modified our statements regarding induction of MICA/MICB expression. Proportions of CD4+ cells in DC-stimulated vs PHA-activated PBLs have been included in a new Supplementary Table (S1).

Comment #2:

The data regarding induction of CT antigens induced by 5-azacytine is of great interest. Do the authors have any data regarding the duration of this alteration in CT protein synthesis post 5-azacytine exposure?

Response:

We now provide new data showing the kinetics of CT-antigen expression in DC-stimulated, 5-aza-CdR-treated T cells. For all donors tested, expression was stable for at least 3 days. This information has been included in the text (p. 6-7) and as a new Supplementary Fig. S4.

Comment #3:

The data presented in Figure 1 do demonstrate augmentation of the generation of CD4+ T-cells. However, what is not clear is whether the induction of CT proteins is differentially observed in CD4 vs. CD8 T-cells. Are there such data? Also, what were the yields of the T-cells after stimulation with the 5-aza treated Th1 cells.

Response:

Data showing expression of CT antigens in CD4+ cells, CD8+ cells and unseparated cultures after treatment with 5-aza-CdR is provided in a new Suppl. Figure (S5). Data showing the efficiency of CD4+ and CD8+ cells in inducing cell proliferation is shown in a new Suppl. Table (S3). The yields of T cells after stimulation are presented in Suppl. Table 8.

Comment #4:

The partial blocking of INF γ production with the W632 antibody shown in suppl. Figure 9 against the 5-azacytine treated CD4+ cells is of interest when compared to the full blocking of cytokine generation in response to breast cancer cells in Suppl Figure 11. Does the residual anti-T cell activity represent autoreaction CD4+ T-cells? Or other possibilities?

Response:

This is an important point, and we agree that incomplete blockage of INF- γ production after contact of isolated CTLs with 5-aza-CdR-treated CD4+ cells could be explained by autoreactivity of 5-aza-CdR-

treated CD4+ cells. Indeed, according to work by Richardson (1986), such autoreactivity is induced in CD4+ cells after treatment with 5-azacytidine. We were aware of this potential complication and therefore decided at an early stage to use 5-aza-CdR-treated cells only for in vitro immunization rather than for direct injection into patients.

We have made a comment on this on p. 9.

Comment #5:

Several specifics regarding the phase I trial need to be specified:

a) Three injections of cells are given: What is the timing of these infusions; weekly, monthly, other? This needs clarification.

Response:

Therapeutic cells were administered at 4-5 week intervals. This is now stated in Results (p. 10) and Patients and Methods (p. 18).

b) What is the timing of the assays measuring lymphocytes “after” infusion in Suppl. Figure 17?

Response:

Total numbers of leukocytes, lymphocytes and neutrophils were measured 1-2 days after injection of cytotoxic lymphocytes. This is now indicated in Results (p. 12), Patients and Methods (p. 19), and the figure legend.

c) Were the 5-aza Th1, enriched APC sensitized T-cells used for treatment all generated at the beginning, as one lot, or were T-cell batches separately generated for each infusion? The latter is suggested from the discussion on page 15, para 1. This is not specified in the text or supplements. If successive lots of T-cells were generated, did the T-cells exhibit the same attributes and specificities? Were the doses given the same?

Response:

Cells were always prepared from freshly drawn blood, as now specified in Results (p. 10), Patients and Methods (p. 18), and footnote to Supplementary Table 8. For individual batches, the cell numbers injected and the proportion of T cells are listed in Supplementary Table 8. Successive batches had

similar phenotypic characteristics (extensive data on this can be made available upon request). Analysis of specificity was not part of the protocol for this phase I trial, and we therefore cannot provide this type of data at this stage.

d) Other than steroids, did the patients receive any other treatment for their GBMs during the trial?

Response:

The patients received no other treatment for their GBMs during the immunotherapy trial, as now specified in Results (p. 10) and Patients and Methods (p. 18).

Comment #6:

It is also difficult to interpret the SPECT/CT results without specifying how many labelled PBL or 5-aza Th1 sensitized CTLs were administered. This needs to be specified.

Response:

We regret that this important information was missing in the initial submission, and have now specified the numbers of labeled cytotoxic lymphocytes and control leukocytes in the Results (p. 10-11).

Comment #7:

The histology of the tumors from the treated patient6 shows macrophages, but no mention is made of T-cells. Were there T-cells in the residual tumor, or in the necrotic lesions?

Response:

We realize that the presentation of the autopsy results from patient #6 was too vague and descriptive, particularly in our use of the term “tumor bed,” which is a neurosurgical expression describing the region of the brain surrounding the tumor. This may have led to the impression that residual tumor was present. In fact, immunohistochemical analysis of tumor, stem-cell and proliferation markers indicated complete tumor eradication. We have rephrased this paragraph to clarify. In addition, we have described the pattern of T cell staining at the previous tumor location (p. 13) and included a new figure showing remaining clusters of T cells in this area (Supplementary Figure 21).

Comment #8:

Given that the authors have been able to characterize the antigenic specificities, in terms of CT antigens, of the T-cells generated, are there any data demonstrating an increase in the frequency of MAGE-10 specific T-cells in the blood of those patients who achieved disease stabilization?

Response:

Analysis of CT antigen specificities was not included in the protocol for this phase I trial. We are, therefore, unable to provide this type of data at this stage.

Comment #9:

The point regarding the genetically engineered T-cells expressing a high avidity MAGE-3 specific TCR that had severe off-target toxicities is a good one, but, in fairness the MAGEA3 TCRs used in that study were not naturally selected but affinity matured to have an avidity in excess of the natural TCR.

Response:

We thank the Reviewer for this clarifying comment. We have taken the liberty to include the Reviewer's wording (p. 18).